# Decreased Fetal Movements: A Sign of Placental SARS-CoV-2 Infection with Perinatal Brain Injury

**DOI:** 10.3390/v13122517

**Published:** 2021-12-15

**Authors:** Guillaume Favre, Sara Mazzetti, Carole Gengler, Claire Bertelli, Juliane Schneider, Bernard Laubscher, Romina Capoccia, Fatemeh Pakniyat, Inès Ben Jazia, Béatrice Eggel-Hort, Laurence de Leval, Léo Pomar, Gilbert Greub, David Baud, Eric Giannoni

**Affiliations:** 1Materno-Fetal and Obstetrics Research Unit, Department Woman-Mother-Child, Lausanne University Hospital, University of Lausanne, 1011 Lausanne, Switzerland; guillaume.favre@chuv.ch (G.F.); leo.pomar@chuv.ch (L.P.); 2Clinic of Pediatrics, Department Mother-Woman-Child, Lausanne University Hospital, University of Lausanne, 1011 Lausanne, Switzerland; sara.mazzetti@chuv.ch (S.M.); Bernard.Laubscher@chuv.ch (B.L.); 3Department of Laboratory Medicine and Pathology, Institute of Pathology, Lausanne University Hospital, University of Lausanne, 1011 Lausanne, Switzerland; carole.gengler@chuv.ch (C.G.); Laurence.DeLeval@chuv.ch (L.d.L.); 4Institute of Microbiology, Lausanne University Hospital, University of Lausanne, 1011 Lausanne, Switzerland; Claire.Bertelli@chuv.ch (C.B.); Gilbert.Greub@chuv.ch (G.G.); 5Clinic of Neonatology, Department Mother-Woman-Child, Lausanne University Hospital, University of Lausanne, 1011 Lausanne, Switzerland; juliane.schneider@chuv.ch (J.S.); eric.giannoni@chuv.ch (E.G.); 6Department of Pediatrics, Réseau Hospitalier Neuchâtelois, 2000 Neuchâtel, Switzerland; 7Department of Obstetrics and Gynecology, Réseau Hospitalier Neuchâtelois, 2000 Neuchatel, Switzerland; romina.capoccia-brugger@rhne.ch (R.C.); Fatemeh.Pakniyat@chuv.ch (F.P.); ines.ben-jazia@rhne.ch (I.B.J.); 8Department of Obstetrics and Gynecology, Hôpital du Valais—Centre Hospitalier du Valais Romand—Site de Sion, 1951 Sion, Switzerland; Beatrice.Eggel-Hort@hopitalvs.ch; 9Midwifery Department, School of Health Sciences (HESAV), University of Applied Sciences and Arts Western Switzerland, 1011 Lausanne, Switzerland; 10Infectious Diseases Service, Department of Internal Medicine, Lausanne University Hospital, University of Lausanne, 1011 Lausanne, Switzerland

**Keywords:** SARS-CoV-2, COVID-19, perinatal, placental, brain injury, fetal movements, neurosonography, MRI

## Abstract

Neonatal COVID-19 is rare and mainly results from postnatal transmission. Severe acute respiratory syndrome coronavirus 2 (SARS-CoV-2), however, can infect the placenta and compromise its function. We present two cases of decreased fetal movements and abnormal fetal heart rhythm 5 days after mild maternal COVID-19, requiring emergency caesarean section at 29 + 3 and 32 + 1 weeks of gestation, and leading to brain injury. Placental examination revealed extensive and multifocal chronic intervillositis, with intense cytoplasmic positivity for SARS-CoV-2 spike antibody and SARS-CoV-2 detection by RT-qPCR. Vertical transmission was confirmed in one case, and both neonates developed extensive cystic peri-ventricular leukomalacia.

## 1. Introduction

Severe acute respiratory syndrome coronavirus 2 (SARS-CoV-2) is causing a major and devastating pandemic, with pregnant women representing a group at increased risk of severe coronavirus disease 2019 (COVID-19) [1,2,3,4], as with other infectious disease [5]. Nevertheless, neonates are relatively spared, with the majority of infants from COVID-19-affected pregnant women exhibiting a favorable short-term outcome [6,7]. SARS-CoV-2, however, can infect the placenta and compromise its function, leading to fetal distress, intrauterine death, or perinatal asphyxia [6,8,9,10,11]. Case reports have described severe neonatal disease in infants from COVID-19-affected mothers, with respiratory failure and/or brain damage [8,9,10,12]. These findings, however, could not be unambiguously attributed to SARS-CoV-2 infection, due to the absence of documentation of vertical transmission and the presence of comorbidities, in particular prematurity. Vertical transmission is proven when the following criteria from the World Health Organization (WHO) are met: evidence of maternal SARS-CoV-2 infection during pregnancy, in utero fetal SARS-CoV-2 exposure, and SARS-CoV-2 persistence or immune response in the neonate [13].

Here, we present two women who developed mild COVID-19 confirmed by nasopharyngeal PCR during their third trimester of pregnancy. Both presented with decreased fetal movements five days after the onset of symptoms requiring an emergency caesarean section (C-section). The two cases of confirmed and strongly suspected congenital SARS-CoV-2 infection were associated with brain damage in neonates.

## 2. Methods

### 2.1. Patients’ Consent and Ethical Approval

We obtained institutional review board approval and written informed consent from both patients.

### 2.2. Sample Collection and Microbiological Investigation

Within minutes of placental extraction by C-section, the fetal surface of the placenta was disinfected and incised with a sterile scalpel, and 2 swabs and biopsies were obtained as previously described [14]. For the first case, we collected cord blood in the sterile surgical field immediately after clamping the umbilical cord, and collected neonatal endotracheal secretions using a sterile procedure. RNA was extracted using a MagNAPure 96 instrument (Roche, Basel, Switzerland), and quantitative SARS-CoV-2 reverse transcriptase–polymerase chain reaction (RT-qPCR) was performed using an automated platform [15,16] on samples from mothers, placentas, and infants. Quantification was performed using calibrated positive plasmid controls and a calibrated SARS-CoV-2 cell culture supernatant [16]. No amniotic fluid samples were collected for SARS-CoV-2 screening.

### 2.3. Placental Examination and In Situ SARS-CoV-2 Detection

Placentas were fixed in 4% buffered formalin. Sampling was performed as previously described [17]. After hematoxylin and eosin staining of paraffin-embedded tissues, we stained samples for immunohistochemical studies with CD68-PGM1, ACE2, and SARS-CoV-2 spike antibodies. We performed in situ detection of SARS-CoV-2 mRNA by RNAScope technology on 4 μm sections from selected formalin-fixed, paraffin-embedded (FFPE) tissue blocks. SARS-CoV-2 detection and quantification by RT-qPCR was performed for both cases, starting from total RNA extracted from 10 µm thick sections of placental FFPE tissue blocks containing foci of chronic intervillositis.

### 2.4. SARS-CoV-2 Genome Sequencing

The CleanPlex SARS-CoV-2 Panel (Paragon Genomics, Hayward, CA, USA) were used according to the manufacturer’s protocol to amplify the SARS-CoV-2 genome from the RNA used for RT-qPCR, as detailed previously [18]. Tiled amplicon libraries were analyzed using a Fragment Analyzer (standard sensitivity NGS, AATI) and quantified with a Qubit Standard Sensitivity NGS dsDNA kit (Invitrogen, Waltham, MA, USA) before sequencing on an Illumina MiSeq (San Diego, CA, USA). We analyzed sequence reads using GENCOV (https://github.com/metagenlab/GENCOV/releases/tag/1.0, accessed on 1 December 2021), a modified version of CoVpipe (https://gitlab.com/RKIBioinformaticsPipelines/ncov_minipipe, accessed on 1 December 2021). SARS-CoV-2 lineages were assigned with pangolin [19].

## 3. Case Description

### 3.1. Case 1

A primiparous 34-year-old previously healthy pregnant woman presented to a regional hospital at 28 + 4 weeks of gestation with chills, fever, myalgia, ageusia, and anosmia. She tested positive for SARS-CoV-2. As the obstetrical examination was unremarkable, she was discharged home the same day, and symptoms resolved rapidly. Five days later, she presented again for reduced fetal movements, confirmed by ultrasound, which motivated transfer to a tertiary center after a first dose of Betamethasone, 12 mg, for fetal lung maturation. Urine and blood tests were normal, with the exception of thrombocytopenia at 63 G/L (first trimester platelet count was in the normal range) and elevated D-dimers (14,860 ng/mL). Due to non-reassuring fetal heart rate (FHR) (Appendix A), absence of fetal movements, and abnormal fetal Doppler (inversed cerebroplacental ratio) suggestive of fetal distress, an emergency C-section without trial of labor and intact amniotic membranes was performed at 29 + 1 weeks’ gestation.

### 3.2. Case 2

A 29-year-old primigravida with gestational diabetes on diet presented to a regional hospital with fever and flu-like symptoms at 31 + 0 weeks’ gestation. She tested positive for SARS-CoV-2 two days after the onset of symptoms and quarantined at home. Five days later, she presented with decreased fetal movements. As the obstetrical examination was unremarkable, the patient was discharged home. Three days later, she was admitted, complaining of absent fetal movements. Abnormal FHR pattern led to an emergency C-section. Basic laboratory tests were within the normal range.

## 4. Results

### 4.1. Maternal Outcomes

Both mothers recovered well after delivery. Case 1 had persistent anosmia and ageusia, and her platelet count and D-dimers normalized spontaneously. Both were discharged home at 5 days after C-section.

### 4.2. Placenta Analysis

The weight of the two placentas were normal (50–70th percentile) for gestational age. Gross examination of cross sections showed massive transplacental changes with trabeculae and lattice-like deposition of fibrin, affecting more than 80% of the total placental volume (Appendix A). Extensive and multifocal chronic intervillositis, characterized by clusters of CD68-positive histiocytes, filled up the intervillous space (Figure 1). Chorionic villi were largely spared from the inflammatory process. The histiocytic intervillous infiltrate was associated with peri-villous fibrin deposition and extensive placental infarction. Intervillous inflammation closely encircled the chorionic villi, and their trophoblastic cells showed a membranous staining for ACE2, as well as an intense cytoplasmic positivity for SARS-CoV-2 spike antibody in both cases. Areas of villi not expressing the SARS-CoV-2 spike protein were not surrounded by CD68-positive histiocytes. SARS-CoV-2 was detected in areas of chronic intervillositis by RT-qPCR in FFPE tissue blocks of both cases. Quantification of viral E gene was 277 copies per reaction for case 1 and 289 copies per reaction for case 2, in the presence of adequate internal MSTN controls. In situ hybridization for SARS-CoV-2 spike protein mRNA visualized the presence of the virus in villous trophoblastic cells in the foci of chronic intervillositis, mirroring the immunohistochemical pattern of expression of SARS-CoV-2 spike antibody (Figure 1).

### 4.3. Neonatal Outcomes

#### 4.3.1. Case 1

A 1370 g (50–75th percentile) female neonate was born at 29 + 3 weeks of gestation with APGAR scores of 4, 8, and 8 at 1, 5, and 10 min, respectively. She developed respiratory distress due to hyaline membrane disease, was intubated 30 min after birth, and received a dose surfactant intratracheally. A complete blood count performed at birth was within the normal range, and blood cultures remained negative (Table 1). She was mechanically ventilated for 13 h and then extubated to nasal continuous positive airway pressure (nCPAP). Umbilical cord blood collected at birth was positive for SARS-CoV-2 (2000 copies/mL). Tracheal secretions collected 11 h after birth were positive for SARS-CoV-2 (1300 copies/mL). Following extubation, additional SARS-CoV-2 PCR tests were negative.

Head ultrasound (HUS) performed on postnatal day 3 identified a right grade II intraventricular hemorrhage and a focal unilateral periventricular hemorrhagic infarction, which evolved to a right frontal porencephalic cyst, progressive ventricular dilatation, and heterogeneous echogenicities throughout the white matter (Figure 2). Bilateral fronto-parieto-occipital cystic peri-ventricular leukomalacia (cPVL) was observed on postnatal day 25. Brain magnetic resonance imaging (MRI) performed on postnatal day 56 (37 + 1 weeks’ postmenstrual age) confirmed extensive bilateral fronto-parieto-occipital cPVL, ependymal hemorrhage sequelae, and moderate ventriculomegaly (Figure 2). At discharge home (postnatal day 75), the infant continued to have an abnormal neurological examination with axial and lower limb hypertonia.

#### 4.3.2. Case 2

A female neonate was delivered at 32 + 1 week’s gestation, weighing 1800 g (50th percentile). APGAR scores were 2, 4, and 5 at 1, 5, and 10 min, respectively. Umbilical cord arterial pH was 6.69. Due to absence of respiratory efforts and bradycardia, the neonate required bag and mask ventilation and chest compressions during the first 5 min after birth. She was then intubated and started on invasive ventilation. Due to hyaline membrane disease, the neonate received intratracheal surfactant. Blood gas showed severe lactic acidosis. She was transferred to a tertiary care neonatal intensive care unit. Laboratory findings are reported in Appendix A.

The neonate fulfilled criteria for perinatal asphyxia with grade II acute hypoxic-ischemic encephalopathy, according to Sarnat score. She developed multiorgan failure, requiring catecholamine treatment, fresh frozen plasma, and platelet transfusions. Persistent pulmonary hypertension was confirmed by echocardiography and required treatment with inhaled nitric oxide for 24 h. She was extubated to nCPAP on postnatal day 4. Intravenous antibiotics, started the day of birth, were stopped after 72 h, as blood cultures were negative. Nasopharyngeal swabs collected at 16 and 76 h following birth and cerebro-spinal fluid collected on postnatal day 5 were negative for SARS-CoV-2 (Table 2).

At 18 h of life, she presented with status epilepticus lasting 3 h that resolved after 3 intravenous doses of midazolam. Recurring electrographic seizures on postnatal day 3 prompted a loading dose of phenobarbitone. The electroencephalogram showed a severely suppressed pattern.

HUS performed on postnatal days 1, 2, and 3 showed a right-sided germinal matrix hemorrhage, cerebral edema, and hyperechogenicity in the fronto-parietal white matter. Brain MRI on postnatal day 7 revealed multiple intraventricular and parenchymal hemorrhages and severe anoxic lesions affecting the white matter and basal ganglia (Figure 3). Hemorrhagic lesions were compatible with asphyxia, coagulopathy, thrombocytopenia, and thrombosis of the straight sinus. She was discharged home on postnatal day 41 (38 weeks postmenstrual age), at which time she continued to have an abnormal neurological examination with limb hypertonia. A repeat brain MRI on postnatal day 55 displayed parenchymal hemorrhagic sequelae, widespread bilateral cPVL, and a non-occlusive thrombus in the straight sinus (Figure 3).

### 4.4. SARS-CoV-2 Sequencing

Genome sequences obtained from nasopharyngeal swabs (case 1 and 2) and a fragment of the placenta (case 2) were attributed to the PANGO lineage B.1.221, a European lineage with increasing prevalence among sequences available in public databases from September 2020 to March 2021, concomitant with the second wave. Genome sequences from both cases only differed by one synonymous mutation. The consensus genome sequences of all three samples are available in GISAID with accession numbers EPI_ISL_2359178 for case 1, and EPI_ISL_2367310 and EPI_ISL_2367312 for case 2.

## 5. Discussion

Both cases illustrate severe neurological injury following a clinical history of decreased fetal movements five days after mild maternal COVID-19. Pregnant women and healthcare providers should be aware that even with non-severe forms of the disease, reduced fetal movements is a sign of potential placental and fetal involvement and should prompt an urgent obstetrical evaluation of fetal well-being.

According to the WHO definition [13], the first case meets criteria for transplacental transmission of SARS-CoV-2. The virus was detected by PCR in the mother–placenta–newborn triad. Vertical transmission through the placenta was supported by direct identification of SARS-CoV-2 on the fetal side of the placenta by in situ techniques (immunohistochemistry, RNA-Scope) corresponding to SARS-CoV-2-induced placental intervillositis. Negative maternal and neonatal SARS-CoV-2 IgG and IgM in case 1 does not rule out an infection in the newborn, as seroconversion can occur within the first 30 days after onset of symptoms [21,22]. The second case remains a suspected vertical transmission. Despite a positive maternal nasopharyngeal PCR and placental detection of the virus, neonatal nasopharyngeal swabs remained negative for SARS-CoV-2, which could be explained by the potential instability of viral RNA [23] or by a rapid viral clearance by the neonate, which occurred in less than two days in the first case. This raises the interesting possibility that preterm newborns can mount an effective immune response against SARS-CoV-2. It is also possible that, despite widespread SARS-CoV-2 placental infection, the virus did not reach the fetus in the second case.

Schwartz et al. [24] identified chronic intervillositis in placentas from SARS-CoV-2-infected maternal-fetal dyads. The inflammatory pattern of chronic intervillositis strongly suggests placental invasion by SARS-CoV-2 and represents a possible mechanism by which the virus can breach the maternal–fetal interface. The identification by in situ techniques of viral particles in the villous trophoblastic cells highly expressing ACE2 may explain the predominance of inflammation in the intervillous space, which differs from chronic villitis caused by other viral agents. Our cases underline the potential for placental infection by SARS-CoV-2 and demonstrate its ability to cause fulminant placental parenchymal destruction, leading to fetal distress within days of mild maternal disease [8,9,10,11,25].

Both preterm neonates developed extensive cPVL. Due to progress in perinatal care, cPVL has become extremely rare [26]. The pathogenesis of cPVL is multifactorial, involving ischemia, inflammation with or without infection, oxidative stress, and excitotoxicity [27]. Given the extent and the type of brain damage observed in our patients, we consider that SARS-CoV-2 is likely to have directly or indirectly caused cPVL. Placental dysfunction occurred in both cases, and led to perinatal asphyxia, hypoxic-ischemic encephalopathy, and severe hypoglycemia in the second case. Yet, these complications do not fully explain the nature and the severity of the brain injury. In adults and children, neurological manifestations have been reported during SARS-CoV-2 infection [28,29,30]. However, viral RNA was detected in the cerebrospinal fluid in a minority of adult patients with neurologic symptoms. In our study, a lumbar puncture performed in case 2 showed no evidence of central nervous system invasion by the virus. Other potential mechanisms for brain injury have been suggested, including systemic inflammation, immune-mediated damage, vasculitis, and thromboembolic events [31]. Our cases do not meet criteria for fetal inflammatory response syndrome [32], and we cannot determine whether vasculitis or thromboembolic events could have contributed to perinatal brain injury.

With a measured rate of 22.8 mutations per year [33], one mutation is expected every two transmissions. The single base difference between both cases suggests a short contact chain, although no epidemiological link could be identified (the two women lived 200 km apart). The limited number of cases does not allow testing for associations between mutations and severity of neonatal disease. It is striking, however, that the same B1.1.221 lineage was involved in both cases presenting with severe neonatal brain injury.

## 6. Conclusions

SARS-CoV-2 can cause severe placental damage and acute fetal distress within days of mild maternal infectious symptoms, leading to extensive cerebral lesions in the infants. The only clinical symptom was maternal perception of decreased fetal movements. Pregnant patients and healthcare professionals should be aware of rare but possibly severe outcomes related to SARS-CoV-2 infection in pregnancy. Information on severe outcomes is the basis of an effective and secure healthcare system, as demonstrated in previous viral crises [34]. Serial HUS should be performed to detect neonatal white matter damage when placental COVID-19 is confirmed. More research is needed to understand the long-term impact of COVID-19 on the developing brain, as well as to confirm whether one of the mutations present in the viral lineage B.1.221 is specifically associated with brain injury or whether this may occur with other variants.

## Figures and Tables

**Figure 1 viruses-13-02517-f001:**
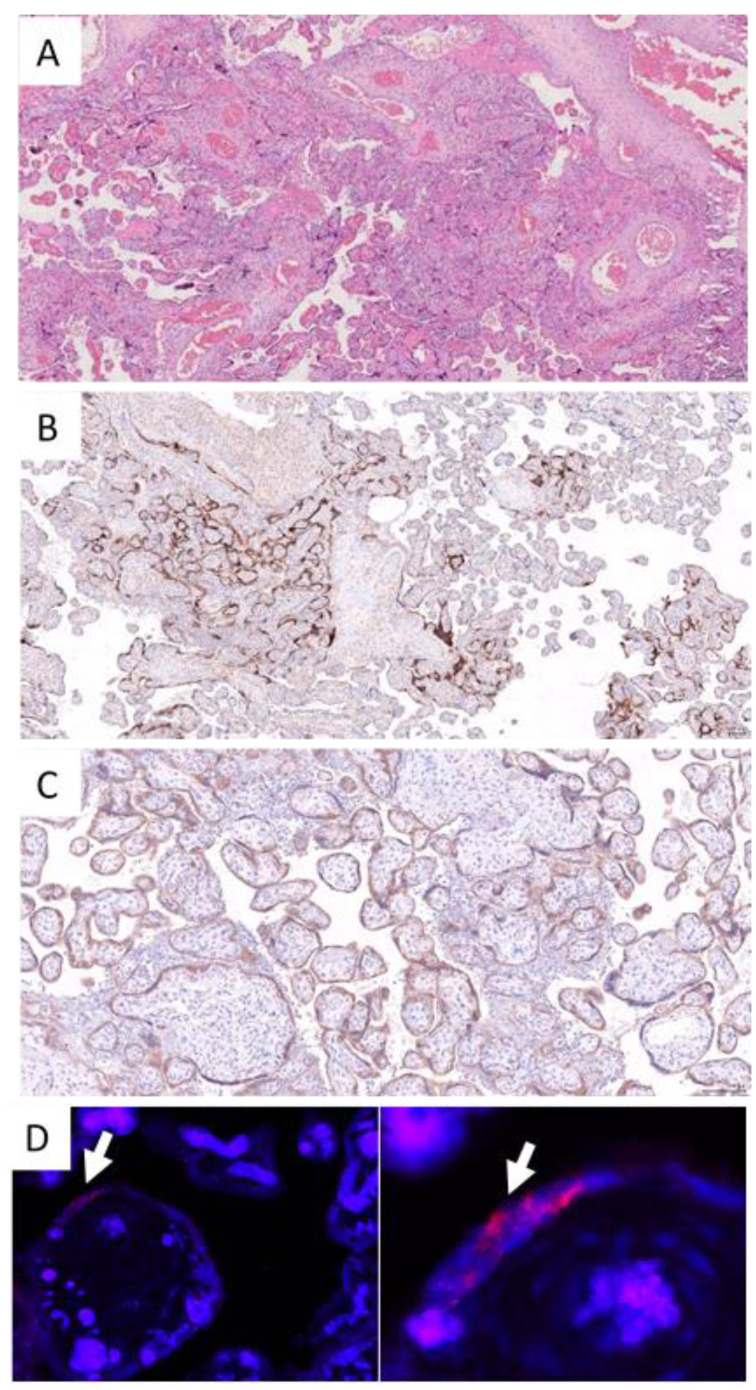
Placental examination—case 1. (**A**) Foci of chronic intervillositis with massive peri-villous fibrin deposition and early placental infarction (hematoxylin and eosin, original magnification ×4). (**B**) Strong cytoplasmic positivity of villous trophoblastic cells with SARS-CoV-2 spike antibody co-localized with the foci of chronic intervillous inflammation (hematoxylin and eosin, original magnification ×4). The following antibodies were used: CD68-PGM1 (Dako Monoclonal Mouse Anti-Human CD68, Clone PG-M1, dilution 1/200), ACE2 (Atlas antibodies, clone CL4035, dilution 1/1000), and SARS-CoV-2 spike antibody (Sino Biological, SARS-CoV Spike S1 Subunit Antibody, Rabbit PAb, Antigen Affinity Purified, dilution 1/250). (**C**) Diffuse and strong membranous ACE2 expression of cytotrophoblastic cells (original magnification ×10). (**D**) In situ hybridization for SARS-CoV-2 spike protein mRNA by RNA scope. In situ detection of SARS-CoV-2 mRNA was performed on 4μm sections from selected formalin-fixed, paraffin-embedded (FFPE) tissue blocks. Actively transcribed SARS-CoV-2 was detected by RNAScope technology (ACDBio, Newark, CA, USA) using a probe specific for the SARS-CoV-2 S protein (2.5VS Probe-V-nCoV2019-S, ACDBio), as previously reported [19]. SARS-CoV-2 detection and quantification by RT-qPCR was performed for both cases, starting from total RNA extracted from 10 µm thick sections of placental FFPE tissue blocks containing foci of chronic intervillositis, using a Cobas z480 instrument (Roche Diagnostics, Basel, Switzerland); one-step RT-qPCR LightCycler^®^ Multiplex RNA Virus Master Mix (Roche Diagnostics); and the following primers: LightMix^®^ Modular SARS, Wuhan CoV E-gene, and LightMix^®^ Modular MSTN extraction control (Roche Diagnostics), as previously reported. The limit of detection (LoD) for E gene was 7 copies per reaction, as previously determined in our laboratory by Probit regression analysis [20].

**Figure 2 viruses-13-02517-f002:**
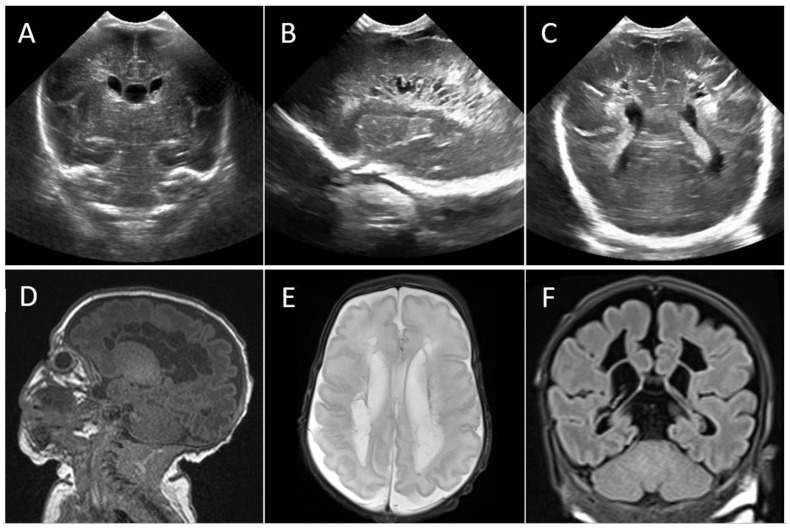
CASE 1: Postnatal brain imaging of the neonates. (**A**) Postnatal day 11 head ultrasound (HUS) coronal view showing a right-sided grade II intraventricular hemorrhage and a focal unilateral periventricular hemorrhagic infarction. Postnatal day 24 HUS with sagittal (**B**) and coronal (**C**) views showing extensive periventricular fronto-parieto-occipital cystic lesions. (**D**) Brain magnetic resonance imaging (MRI) on postnatal day 56 (37 + 1 weeks postmenstrual age) with sagittal T1-weighted, (**E**) axial T2-weighted, and (**F**) coronal T2-FLAIR (fluid-attenuated inversion recovery) images confirming severe periventricular cystic leukomalacia, sequelae of germinal hemorrhage, and moderate ventriculomegaly.

**Figure 3 viruses-13-02517-f003:**
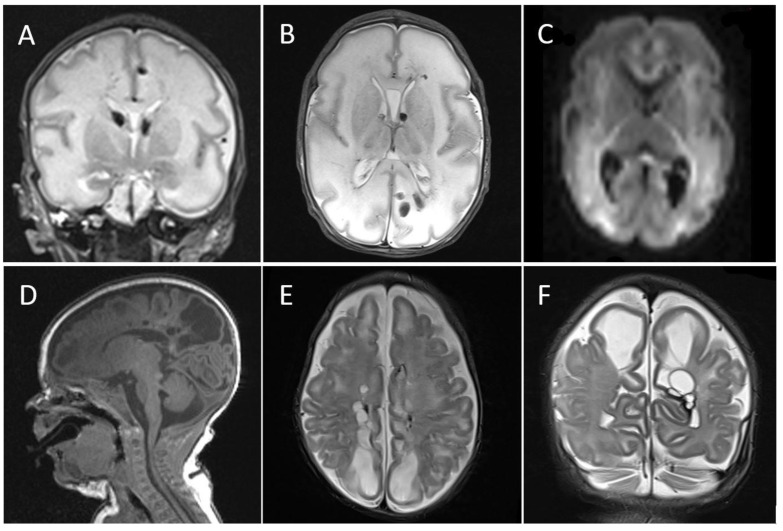
CASE 2: Postnatal brain imaging of the neonates. Postnatal day 7 MRI with (**A**) coronal, (**B**) axial T2-weighted, and (**C**) axial diffusion-weighted images showing germinal, intraventricular, and multiple parenchymal hemorrhages, as well as severe anoxic lesions affecting the white matter bilaterally predominantly in parieto-occipital regions and the basal ganglia. (**D**) Postnatal day 55 (40 weeks postmenstrual age) MRI with sagittal T1-weighted, (**E**) axial, (**F**) and coronal T2-weighted images showing transformation of the anoxic lesions to bilateral cystic periventricular leukomalacia, predominantly in parieto-occipital lobes and sequelae of germinal matrix and fronto-parietal white matter hemorrhage.

**Table 1 viruses-13-02517-t001:** Case 1: neonatal, maternal, and placental microbiology.

	Before Birth	Birth	H11	H36	H48	DOL 4	DOL 7
**Newborn**							
Cord blood PCR	-	POSITIVE ^1^	-	-	-	-	-
Tracheal secretion PCR	-	-	POSITIVE ^2^	-	-	-	-
Nasopharyngeal swab PCR	-	-	-	Negative	-	-	-
Serum PCR	-	-	Negative	-	Negative	-	Negative
Serology IgG	-	-	-	Negative	-	-	-
Serology IgM	-	-	-	Negative	-	-	-
Blood culture	-	Negative	-	-	-	-	-
**Mother**							
Nasopharyngeal swab PCR	POSITIVE	-	-	-	-	-	-
Serology IgG	-	-	-	-	-	POSITIVE ^3^	-
**Placenta**							
Placenta swab PCR IgM	-	POSITIVE ^4^	-	-	-	-	-
SARS-CoV-2 identification	-	POSITIVE ^5^	-	-	-	-	-

DOL: day of life; ^1^ 2.0 × 10^3^ copies/mL; ^2^ 1.3 × 10^3^ copies/mL; ^3^ 99.6 (ratio), negative if <4; ^4^ 2.8 × 10^7^/mL; ^5^ RNAscope ISH assay.

**Table 2 viruses-13-02517-t002:** Case 2: neonatal, maternal, and placental microbiology.

	Before Birth	Birth	H5	H16	DOL 3	DOL 4	DOL 5
**Newborn**							
Nasopharyngeal swab PCR	-	-	-	Negative	Negative	-	-
Blood culture	-	-	Negative	-	-	Negative	Negative
Cerebrospinal fluid PCR	-	-	-	-	-	-	Negative
**Mother**							
Nasopharyngeal swab PCR	POSITIVE	-	-	-	-	-	-
**Placenta**							
Placental swab PCR	-	POSITIVE ^1^	-	-	-	-	-
SARS-CoV-2 identification	-	POSITIVE ^2^	-	-	-	-	-

DOL: day of life; ^1^ 3.0 × 10^6^/mL; ^2^ RNAscope ISH assay.

## Data Availability

Not applicable.

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
