# Peer review of "Decreased Fetal Movements: A Sign of Placental SARS-CoV-2 Infection with Perinatal Brain Injury"

_viruses, 2021, doi:10.3390/v13122517_

Round 1

Reviewer 1 Report

Thank you for the opportunity to review the article Favre G. et al.
Decreased Fetal Movements: a Sign of Placental Sars-Cov-2 Infection
With Perinatal Brain Injury.
Nowadays, in the Covid-19 era, any report may influence the
therapeutic process used by doctors.
Therefore, I believe that the proposed report is very timely.
So far, few studies show the effects of Covid-19 in newborns.
Therefore, I believe that the work is worth posting after some
minor revision.
However the paper is very interesting, some changes are required.

2.1 - situation quite obvious, I do not think that this information
is relevant for the manuscript - until the author's decision
2.2 - w
as the presence of the virus in the amniotic fluid assessed
during the treatment phase?
3.1 -
what tests were performed 1st and 2nd? Thrombocytopenia was before?
3.2 - what tests were performed 1st and 2nd?
Conclusion -
Whether Covid-19 was the cause of brain damage in newborns or maybe it would be better to describe the first pregnancy woman in
particular
to get rid of the doubts - until the author's decision

Author Response

REVIEWER 1 COMMENTS

Thank you for the opportunity to review the article Favre G. et al. Decreased Fetal Movements: a Sign of Placental Sars-Cov-2 Infection With Perinatal Brain Injury. Nowadays, in the Covid-19 era, any report may influence the therapeutic process used by doctors. Therefore, I believe that the proposed report is very timely. So far, few studies show the effects of Covid-19 in newborns. Therefore, I believe that the work is worth posting after some minor revision. However the paper is very interesting, some changes are required.

2.1 - situation quite obvious, I do not think that this information is relevant for the manuscript - until the author's decision

As suggested, we removed the section regarding COVID-19 routine precautions in section 2.1 and thus renamed the section as following:

Line 61 - “2.1. Patients’ consent and ethical approval”

2.2 - was the presence of the virus in the amniotic fluid assessed during the treatment phase?

We thank the reviewer fort this relevant point. Unfortunately, amniotic fluid samples were not collected before and during delivery. We then modified the methods section with the following text:

Line 73 - “No amniotic fluid samples were collected for SARS-CoV-2 screening”

3.1 - what tests were performed 1st and 2nd? Thrombocytopenia was before?

Thank your for that important point.

Both pregnancies were physiological except a gestational diabetes on diet for case 2. Both patients had routine pregnancy follow-up according to WHO guidelines with routine blood tests and ultrasound in the first and second trimester (https://www.who.int/publications/i/item/9789241549912/).

The first case had a normal blood count, including normal platelets, at the first trimester examination. We then added the following:

Line 102 -  […] “at (63 G/L (first trimester platelet count was in the normal range) […]

3.2 - what tests were performed 1st and 2nd?

We answered to the comment in comment 3.1

Conclusion -Whether Covid-19 was the cause of brain damage in newborns or maybe it would be better to describe the first pregnancy woman in particular to get rid of the doubts - until the author's decision

We agree with the reviewer’s comment. Unfortunately, we do not have all the tests we would need to confirm vertical transmission in case 1, as we do not have an immediate neonatal cord blood sample for SARS-CoV-2 screening.

Reviewer 2 Report

I read with great interest the paper. I find it well wrote and with very good idea research. Only minor suggestion for a very excellent and well presented paper.

  1. Introduction: updata data on burden SARS CoV2 at the day of resubmission. Also introduce the experience that infectious diseases during pregnancy are most danguerous (as for malaria or other viral infection, see and cite Maternal caesarean section infection (MACSI) in Sierra Leone: a case-control study. Epidemiol Infect. 2020 Feb 27;148:e40)
  2. Methods and results: very clear and well presented
  3. Discussion: excellent
  4. Conclusion: give some proposal both clinical and in global health perspectives that came from your excellent paper. Also the central role of pregnancy during pandemic. Other previous experience showed how during epidemic (see and cite Impact of Ebola outbreak on reproductive health services in a rural district of Sierra Leone: a prospective observational study. BMJ Open. 2019 Sep 4;9(9):e029093), maternal services had a worste disruption with a worste outcome for pregnant and baby

congratualtions for your great paper

Author Response

I read with great interest the paper. I find it well wrote and with very good idea research. Only minor suggestion for a very excellent and well presented paper.

Introduction: updata data on burden SARS CoV2 at the day of resubmission. Also introduce the experience that infectious diseases during pregnancy are most danguerous (as for malaria or other viral infection, see and cite Maternal caesarean section infection (MACSI) in Sierra Leone: a case-control study. Epidemiol Infect. 2020 Feb 27;148:e40)

We thank the reviewer for his comment. We then add the following sentence and reference to the manuscript:

Line 43 – […] with pregnant women representing a group at increased risk of severe coronavirus disease 2019 (COVID-19) [1–4], as fwith other infectious diseases [5].

5- Maternal caesarean section infection (MACSI) in Sierra Leone: a case-control study. Epidemiol Infect. 2020 Feb 27;148:e40)

Methods and results: very clear and well presented

Discussion: excellent

We are honored of the reviewer’s comment.

Conclusion: give some proposal both clinical and in global health perspectives that came from your excellent paper. Also the central role of pregnancy during pandemic. Other previous experience showed how during epidemic (see and cite Impact of Ebola outbreak on reproductive health services in a rural district of Sierra Leone: a prospective observational study. BMJ Open. 2019 Sep 4;9(9):e029093), maternal services had a worste disruption with a worste outcome for pregnant and baby

congratualtions for your great paper

We thank the reviewer for this great idea. We then added to the conclusion:

Line 340 - Pregnant patients and health care professionals should be aware of rare, but possible severe outcomes related to SARS-CoV-2 infection in pregnancy. Information on severe outcomes is the basis of an effective and secure health care system, as demonstrated in previous viral crises [33].

33- Quaglio G, Tognon F, Finos L, et al. Impact of Ebola outbreak on reproductive health services in a rural district of Sierra Leone: a prospective observational study. BMJ Open. 2019;9(9):e029093. Published 2019 Sep 4. doi:10.1136/bmjopen-2019-029093